# Polymer-Based Nanofiber–Nanoparticle Hybrids and Their Medical Applications

**DOI:** 10.3390/polym14020351

**Published:** 2022-01-17

**Authors:** Mingxin Zhang, Wenliang Song, Yunxin Tang, Xizi Xu, Yingning Huang, Dengguang Yu

**Affiliations:** 1School of Materials and Chemistry, University of Shanghai for Science and Technology, Shanghai 200093, China; 203613006@st.usst.edu.cn (M.Z.); 213353179@st.usst.edu.cn (Y.T.); 192432632@st.usst.edu.cn (X.X.); 1935040104@st.usst.edu.cn (Y.H.); 2Shanghai Engineering Technology Research Center for High-Performance Medical Device Materials, Shanghai 200093, China

**Keywords:** polymer composites, nanoparticle, polymer blends, medical applications, electrospinning

## Abstract

The search for higher-quality nanomaterials for medicinal applications continues. There are similarities between electrospun fibers and natural tissues. This property has enabled electrospun fibers to make significant progress in medical applications. However, electrospun fibers are limited to tissue scaffolding applications. When nanoparticles and nanofibers are combined, the composite material can perform more functions, such as photothermal, magnetic response, biosensing, antibacterial, drug delivery and biosensing. To prepare nanofiber and nanoparticle hybrids (NNHs), there are two primary ways. The electrospinning technology was used to produce NNHs in a single step. An alternate way is to use a self-assembly technique to create nanoparticles in fibers. This paper describes the creation of NNHs from routinely used biocompatible polymer composites. Single-step procedures and self-assembly methodologies are used to discuss the preparation of NNHs. It combines recent research discoveries to focus on the application of NNHs in drug release, antibacterial, and tissue engineering in the last two years.

## 1. Introduction

COVID-19 has contributed to a worldwide healthcare crisis resulting in several hundreds of thousands of deaths over recent years and complications that can lead to severe pneumonia [1,2]. In synergy with Polymer Engineering, Bioengineering, and Advanced Manufacturing, the mRNA vaccine was successfully developed and blocked the further spread of the virus on a large scale [3,4]. The mRNA-1273 vaccine approved for use under urgent circumstances has the potential to be unsafe for delivery, being inherently unstable while the large density of negative charges makes it difficult to cross cell membranes [5,6], and the latest research uses polymer-based composites as carriers for effective vaccine delivery [7].

In medical applications, polymers with superior biocompatibility are often used to not produce antibody reactions or even stimulate inflammation in contact with the human body [8,9,10]. Synthetic polymers include: polycaprolactone (PCL), poly (vinyl alcohol) (PVA), polylactide (PLA) and poly (lactic-co-glycolic acid) (PLGA) [11,12]; natural polymers include: Chitosan (polysaccharide), cellulose (polysaccharide), gelatin (protein), silk proteins (proteins) [13], etc. In biomedical applications, natural polymers are often perceived as relatively safe, biocompatible and degradable polymers [14]. Cellulose is a linear polysaccharide, the main component of the cell wall, which is currently the most used in biological applications from bacteria [15]. Taokaew et al. composite nanofibers based on bacterial cellulose with Garcinia mangostana peel extract exhibited potent bacterial inhibition against Gram-positive bacteria while treating the membranes with MCF-7 breast cancer cells for 48 h, only 5% of the cancer cells remained viable [16]. The protein class of silk fibroin, unlike other natural polymers, has outstanding elasticity and strength, which is of interest in tissue engineering applications [17,18].

The advantages of synthetic polymers over natural polymers are their stability, superior mechanical properties and degradability. PCL is an aliphatic semicrystalline polymer with hexanoic acid formed by esterolysis [19], which can be excreted through the digestive system, thus making it biocompatible, biodegradable, non-toxic and popular in medical applications, such as drug delivery, wound plug dressings and stents. The use of PCL scaffolds for tissue engineering has been reported in detail by Janmohammadi et al. [20]. Zavan and co-works prepared a bilayer fibrous scaffold with the outer layer of PCL exhibiting the best in vitro response to fibroblasts and weak adhesion to cellular tissues due to the hydrophobicity of PCL to cellular tissues, while the inner layer used gelatin-modified PCL to significantly promote gene expression in endothelial cells [21].

Nanoscale polymers have a wide range of uses in various fields, especially in medical applications, where the electrospinning process (Figure 1) is one of the most convenient ways to prepare nanofiber membranes (1–100 nm) directly and continuously [22,23,24,25,26]. Laboratory electrospinning setup generally has three components: a high voltage electrostatic generator (1–30 kv), a spinning head and a collector (opposite charge target) for the process of producing various continuous nanofiber assemblies with controlled morphology and dimensions from polymer solutions or melts under a high voltage electric field [27,28,29]. The development of the electrospinning (ES) process was a somewhat difficult journey. In the 16th century, William Gibert discovered a conical droplet formed when charged amber approached water droplets. Micro-droplets are also ejected from the conical droplets. The beginning of electrospinning technology is recognized to be in 1934, when Anton Formalas invented a device to prepare polymer fibers by electrostatic force action, presenting for the first time how a polymer solution could form the jet between electrodes. Subsequent years of intermittent patent publication have not attracted much attention from researchers. In 1990, Reneker’s research team at the Acolon University, USA, went deeper into the electrospinning (ES) process and applications, with ES of various organic polymer solutions into fibers, ES process is rapidly becoming a research hotspot, entering a new era of vigorous development [30,31].

To prepare the desired nanofibers, the electrospinning process usually requires setting parameters, the main influencing factors being the system (nature of the polymer, solution viscosity, conductivity and surface tension) [32], process parameters (voltage, receiving distance and flow rate) [33] and environmental factors (temperature and humidity) [34]. Specific effects on electrospun fiber morphology are shown in Table 1.

Synthetic polymers have modifiable mechanical properties, but the hydrophobic nature contributes to an absence of cell adhesion and the onset of inflammation, which is usually modified by natural polymers. One problem that most natural polymers suffer from is the deficiency of certain mechanical properties [48]. To combine the advantages of various polymers and overcome their limitations, the desired properties have been achieved using blends of these polymers instead of using single polymers by optimizing the ratio between the components of the blend. During drug release, the hydrophobic properties of the polymer seriously affect the release performance of the loaded drug. Huo and co-workers used PCL blended with gelatin to effectively improve the hydrophobic properties of the composite fiber. Drug release demonstrated that the increased PCL content made the composite fiber more hydrophobic, which belonged to the slow release of artemisinin (ART) and enhanced the therapeutic effect [49]. Chen et al. improved the defect of gelatin’s poor hemostasis ability for large wounds by co-mixing sodium alginate, which can form a gel on the wound surface, and achieved hemostasis of rat wounds in only 1.53 min [50].

Nanofiber membranes are prepared by the ES process of polymers, and the application scenario is constrained to fiber scaffolds [51,52,53]. Researchers are not satisfied with this single role, so various functional nanofiber membranes have been prepared by ES in combination with multifunctional materials, among which NNHs show great potential to combine the advantages of nanoparticles with the properties of polymers. Additionally, the is a wide range of application scenarios [51,52,53,54,55,56,57,58]. Of interest is that as with decades of research, the number of electrospun nanoparticle articles published occupies half the number of electrostatic spinning (Figure 2). It shows the pivotal position of NNHs.

Multiple nanoparticles have been successfully prepared into nanofibers, which mainly include: metal [59], metal oxide [60] and polymer nanoparticles (NPs) [61,62]. The presence of NPs in polymer solution is uniformly dispersed and will be randomly distributed on the surface or inside the nanofibers (NFs). Therefore, the co-blending of NPs with polymer solutions is the most common form. During the process of electrospinning the mixture into fibers, the homogeneity of NPs in the polymer solution and the interfacial interaction between nanoparticles and polymer solution significantly affect the performance of NNHs. To attenuate the effects of the above factors, the NPs can be mixed by physical mixing methods (stirring and sonication), by dissolving the NPs and polymer in different solvents separately or adding a certain amount of surface activator. The electrospun fiber precursor working solution can effectively reduce the phenomenon of nanoparticle agglomeration and even blockage of the spinning heads after mixing uniformly [63,64,65]. Secondly, other processes combined with ES technology can also effectively load NPs into NFs. For instance, the modification of NFs by plasma technology. This methodology promotes the interaction between NPs and easily loads NPs on the nanofiber surface [66]. The electrospray [67] or magnetron sputtering [68] technology allows the uniform spraying of NPs on the fiber surface. The single-fluid ES process requires mixed solutions with a certain viscosity and dispersion. Otherwise, a non-spinnable or agglomerative phenomenon will occur. Multi-fluid dynamics have been studied for many years. The design of the spinneret is especially important. The morphological structure of the nanofibers is similar to the spinneret structure [69]. Multi-fluid dynamics ES processes often require only one fluid available for spinning to form nanofibers [70]. Zheng et al. used TiO_2_ suspension as the sheath layer and PEO as the core layer, which perfectly avoided the enrichment of TiO_2_NPs using a modified coaxial electrospinning process, and the uniform TiO_2_NPs on the nanofiber surface enhanced the absorption of UV light [71]. The NPs are loaded in nanofibers based on the above single-step process, and if the NPs are not well dispersed in the working solution, many self-assembly strategies are available to grow NPs in the fibers by initiation effects in the precursor solution, including: in situ synthesis [72], hydrothermal-assistance [73] and calcination [74].

NNHs prepared by combining the characteristics of biocompatible polymers and multifunctional nanoparticles are already practical in medical applications [75,76]. The objective of this review is to investigate the preparation of nanofiber and nanoparticle hybrids in-depth by combining different processes with ES; concentrate on the most recent breakthroughs of NNHs in the medical area (Figure 3); and provide clear concepts for research workers in material preparation and application.

## 2. The Methods for Creating Polymer-Based Nanofiber-Nanoparticle Hybrids (NNHs)

### 2.1. Overlapping of Electrospinning and Other Techniques

ES is a top-down molding process for the direct and continuous preparation of nanofibers. NNHs in the preparation of NPs in the spinning precursor solution, due to a certain stability, will not form a homogeneous solution in organic solvents. In addition to the large-scale manufacturing of NNHs, other technologies are often used in further combination with ES.

AgNPs in PCL nanofibers suffer from inhomogeneous dispersion and irregular ion release. To address this problem, Valerini and co-workers employed the magnetron sputtering technique to sputter Ag targets onto the surface of PCL nanofibers at a low power of 500 w. SEM images of PCL NFs after Ag magnetron sputtering show higher contrast. Composite NFs with 22 nm size AgNPs were deposited on the surface. Although the Ag content was only 0.1 wt.%, the PCL-Ag nanofibers showed a faster and stronger antibacterial effect by depositing all AgNPs on the nanofiber surface by magnetron sputtering technique [77]. Immersion of nanofibers (NFs) in NPs suspension loaded with NPs is the simplest approach; however, significant rejection occurs due to the different hydrophilic properties of NPs and NFs, leading to ineffective results. Liu et al. introduced oxygen-containing polar functional groups to the surface of PLLA membranes by a facile plasma technique. Negatively charged pPLLA membranes, impregnated in AgNPs solution, were well aggregated with NPs on the fiber surface driven by electrostatic interactions [78]. Electrospray, a sister technology to electrospun fibers, is commonly employed to generate NPs [79]. Fahimirad and co-workers sprayed chitosan NPs loaded with curcumin onto the surface of PCL/chitosan/curcumin nanofibers by electrospray, which showed no significant change in nanofiber diameter. While effectively promoting the swelling ability and degradation rate of nanofibers, the composite fibers healed 98.5% of the wounds infected by MRSA through wound closure experiments [80].

### 2.2. Encapsulating Nanoparticles in the Working Fluids

The presence of NPs necessarily affects the spinnability and composite fiber morphology during the preparation of NNHs. NPs appear agglomerated in the working solution, and needle blocking may occur during ES, or the NPs will not appear uniformly on the fibers. These issues can lead to reduced mechanical properties and loss of functional sites in composite fiber membranes. There are three ways to deal with them to avoid a similar situation: (1) disperse NPs uniformly in the working solution by physical methods such as stirring; (2) separately dissolve NPs and polymers in different solvents; (3) modify them using surfactants to promote uniform dispersion. With different natures of nanoparticles and polymer solutions in the mixing process, there is a large interfacial force, and the mixture can be added to a quantitative amount of surfactant, promoting the orderly arrangement of nanoparticles no longer agglomeration. Karagoz and co-workers took this approach by adding a quantitative amount of Triton x-100 (surfactant) to a mixture of ZnO nanorods and DMF, sonicated for 20 min, then added the polymer under rapid stirring until complete dissolution. Fiber diameters were uniformly distributed, and no beads were observed [81].

### 2.3. Formation of NNHs in the Single-Step Process

With a precursor working solution using suitable solvent and proper dispersion, the NPs are uniformly dispersed in the working solution, which requires only a single-step process to form NNHs by adjusting the ES process parameters (Figure 4A). Lopresti et al. incorporated silica (AS) or clay (CLO) NPs into 10% PLA solution and gathered them by a cylindrical grounded drum at 15 KV. The phenomenon of particle aggregation or clustering was evident with the increase in NPs content of the composite fibers (Figure 4E). Moreover, the composites with a certain number of added NPs exhibited brittleness, but bone cells diffused faster on the composites with higher particle content [82]. Abdelaziz and co-workers encapsulated silver nanoparticles (AgNPs) and hydroxyapatite (HANPs) in PCL/CA solution. Interestingly, the tensile stress of the composite fibers loaded with 10% HANPs was up to 3.39 MPa, well above that of pure PCL/CA nanofibers, but then the tensile stress of the composite fibers with 20% HANPs dropped sharply to 2.22 MPa, perhaps because the mechanical property change of NNHs with the addition of nanoparticles is a parabolic. This interesting phenomenon could contribute to the reference of optimizing the relationship between NNHs proportioning and mechanical properties [83].

The polymer solution and NPs are divided into different syringes to eliminate the agglomeration of NPs inside the fibers, which are encapsulated in the fibers by coaxial electrospinning or electrospray. Navarro Oliva and co-workers prepared core-shell-structured NNHs by the coaxial electrostatic spinning technique using an Fe_3_O_4_ solution as the core layer and a PVDF polymer solution as the shell layer. As shown in Figure 4B, TEM images showed that the NPs were uniformly dispersed inside the fibers without agglomeration [84]. Combining both electrospinning and electrospraying to prepare NNHs is also a convenient method. These two processes are carried out simultaneously, and NPs can be uniformly distributed in the fiber layer (Figure 4C).

The electrospinning process (ESP) has evolved from single-fluid to multi-fluid as research has progressed [86]. Yu’s research team successfully prepared nanofibers with obvious core-shell structures and smoother fiber surfaces by modified tri-axial ESP and achieved intelligent three-stage controlled drug release [87,88]. Pursuing a simple and convenient preparation method, the main preparation method of NNHs is currently the uniaxial ES process. Integrating the preparation of NNHs with more sophisticated ES techniques is a challenge. The incorporation of NPs somewhat reduces the capability of NFs. There are two obvious drawbacks: (1) The addition the NPs reduces the interactions between polymer chains and decreases the mechanical properties of NNHs. (2) NPs are wrapped in fibers and cannot be functionalized. However, Radacsi et al. avoid the defect that NPs are encapsulated by nanofibers and cannot realize the specific surface area enhancement by using cesium dihydrogen phosphate (CDP) for spontaneous nucleation along the solute growth in porous nanofibers; the moist water in the air acts as a solute transport fast channel. Thus, NPs are not only grown inside the fibers but also are uniformly dispersed on the fiber surface. This strategy perfectly avoids the defect that NPs are encapsulated by nanofibers and cannot realize the specific surface area enhancement (Figure 4D) [85].

### 2.4. The Nanoparticles from Nanofibers through Molecular Self-Assembly

NPs are formed spontaneously by self-assembly strategies [89,90], combining this scheme with ESP, based on post-processing to form NPs spontaneously in fibers, such as in situ synthesis, hydrothermal-assistance and calcination.

#### 2.4.1. In Situ Synthesis

The smooth surface of electrospun fibers and certain stability of the polymer, NPs cannot be perfectly and uniformly attached to the fiber surface by electrostatic adsorption or functional group action. The fiber impregnated with a ligand solution instigates bonding using a reducing agent or other reduction in the composite material to form NPs [91,92]. Lv et al. cross-linked potato starch as a polymer after forming starch fiber mats immersed in AgNO_3_ solution, and Ag^+^ was reduced to AgNPs by heating at 60 °C protected from light. The average particle size of AgNPs increased with the concentration of AgNO_3_, but the particle size decreased significantly and agglomerated together when the concentration reached 100 mg/mL [93]. The technology is also applied to MOF-based nanofibers. Li and co-workers synthesized PW12@UiO-66 crystals in situ on PAA-PVA nanofibers. The surface of the fibers was completely covered by crystals at 12 min of growth, and the MOF crystals were enriched on the surface of NNH but with the disadvantage of random arrangement [94]. Lee and co-workers adopted a hydrodistillation-induced phase separation method to form dense cavities on the surface of PLA nanofibers, followed by uniform growth of ZIF-8 crystals on the porous fiber surface (Figure 5A [95]). Using COF materials with similar properties to MOF, Ma et al. adopted soaking PAN membranes in COF material precursor solutions, where Schiff base condensation occurred at room temperature to form COF spherical particles, and PAN@COF composite fibers showed excellent thermodynamic stability at 300 °C (Figure 5B) [91]. The NNHs prepared by this method, with dense particle aggregation, provide a sufficient number of sites of action that may provide a good solution for the adsorption of tissue fluids or other substances. Similar results from in situ synthesis can be found in Figure 5C [96] and Figure 5D [97].

#### 2.4.2. Calcination

Calcination is another effective method to prepare NNHs. Precursor fibers are prepared by the sol–gel method in combination with ESP, and calcination forms nanoparticles in the fibers [98]. Xie fabricated PVP composite fibers, which were maintained at 800 °C for 30 min, and iron-cobalt (FeCo) alloy nanoparticles (30–50 nm) were uniformly distributed in CNFs (150–300 nm), which achieved electrocatalytic degradation of antibiotics in wastewater [99]. Ding’s research team proposed the use of ball-milling precursor sols to form homogeneous nuclear to precisely control crystal nucleation and growth for the purpose of grain refinement to avoid problems, such as the appearance of impurity phases and crack expansion during the preparation of flexible chalcogenide LLTO nanofibers (Figure 5D). Meanwhile, the soft grain boundary is constructed by adopting 200–900 °C stage calcination, which shows the excellent mechanical properties of flexible electronic fiber films based on perovskite ceramic oxide [92]. Nanofibers obtained by unusual calcination are brittle and inflexible. To reduce such a situation, Shan et al. formed porous structures using the template method with sacrificial PAA heated at 700 °C for 2 h to modulate the PAA concentration driving phase separation during calcination stage pyrolysis (Figure 5C) [91]. Similarly, Li and colleagues inherited the MOF structure in ZIF/PAN-Ni-15 composite nanofibers at the sacrifice of ZIFNPs by calcining at 700 °C for 2 h, followed by annealing to form a rosary morphology in the fibers [74].

#### 2.4.3. Hydrothermal-Assistance

Hydrothermal-assistance is a common method to prepare a variety of inorganic oxide crystals, which is based on the conditions of low temperature and isobaric pressure. The uniform distribution of substances avoids the appearance of impurity phases by means of an aqueous medium, obtaining nanoparticles (such as spheres, cubes, flowers, etc.) with diverse morphologies [100,101,102]. The combination of hydrothermal and ESP for the preparation of multilayer heterostructures provides superior performance at a low cost and is green, easy to operate and well received by researchers. Küçük et al. immersed TiO_2_ fibers (100–200 nm) in an alkaline Ba (OH)2·8H_2_O solution by a simple hydrothermal reaction, which unexpectedly transformed the composite fibers into tetragonal crystals, demonstrating that metal oxide nanofibers can also be precursors for the preparation of BaTiO_3_ crystals [103]. Mukhiya and colleagues grew MOF materials on PAN-prepared carbon nanomats (350 nm). First growth of ZIF-67 crystals was attributed to the hydrothermal reaction of cobalt carbonate hydroxide with 2-methylimidazole. The synthesis of ligands by deprotonated 2-methylimidazole with Co^2+^ was responsible for the crystals’ second growth. The composite fiber membrane has high specific capacity and excellent service life, with strong advantages in energy storage applications [104]. In terms of the same application, Poudel et al. prepared PAN/PMMA nanofiber membranes using a coaxial ESP and sacrificed the internal PMMA after carbonization to obtain 3D hollow carbon nanofibers (3DHPCNF). By adopting two hydrothermal methods, Fe_2_O_3_ was first synthesized on the fiber surface, and then this was used as a precursor to hydrothermally generate ZMALDH@Fe_2_O_3_/3DHPCNF with ternary metal salts in an autoclave. The LDH lamellar structure was thinned by changing the content of Zn, and the transformation of nanosheets to nanowires was found at high Zn^2+^ content, which is a novel top-down way to obtain ternary LDH-electrospun hollow carbon nanofibers, providing a new idea for subsequent design [105].

## 3. The Biomedical Applications of Nanofiber-Nanoparticle Hybrids (NNHs)

Electrospun fibers, which are based on biomaterials, have been developed in biomedical fields such as tissue scaffolds for a long time, but the single role is not enough for sophisticated practical applications. Researchers have combined multi-functionalized NPs with NFs to develop innovative materials for drug delivery, antibacterial and tissue engineering.

### 3.1. Drug Delivery

Drug delivery systems (DDSs) release drugs through specific control devices and certain doses to achieve the purpose of enhancing the immunity of the body or treating diseases [106,107]. It is difficult to advance medicine without the use of medications. While drug activity is crucial, the method of administration is more so. Pharmacokinetics (PK), duration of action, metabolism and toxicity are the key elements that influence drug delivery [108]. NNHs inherit the excellent biocompatibility and high specific surface area of electrospun fibers but also the function of NPs, which can load different drugs into NNHs for a precise controlled release [109,110]. The advantages of NPs are their high specific surface area, superior bioavailability and functionalization. Similarly, in practical drug delivery applications, nanofibers can be electrospun with biocompatible polymers, resulting in minimal harmful effects on humans [111]. Tuğcu-Demiröz and co-workers employ ionic gelation to load benzydamine onto chitosan NPs. Composite NPs were embedded in NFs and hydrogels, respectively. On the one hand, NFs have a large specific surface area. The medicine, on the other hand, diffuses across a short distance in the fiber. Benzydamine released 53.03% in the composite fiber system after 24 h. Because of the strong polymer matrix, the hydrogel system only released 15.09% of benzydamine. This is insufficient for the required drug concentration for vaginal infections [112]. In the process of release, benzydamine enters the nanofiber after the initial release of chitosan nanoparticles, and the secondary release can achieve a slow release and prolong the action time.

For the mechanism of release of NNHs-loaded drugs: (1) one drug in two phases: the drug is loaded on nanoparticles that need to cross the barrier between nanoparticles and nanofibers during the release to achieve the effect of slow release. (2) Dual drug biphasic: two drugs are loaded in separate materials with different properties to achieve the required effect by different types of release mechanisms.

#### 3.1.1. One-Drug Biphasic

One-drug biphasic is an advanced therapeutic approach to chronic diseases by precisely designing the carrier to act rapidly in the first burst and then to work longer with chronic illness. For example, He and co-workers used ethylcellulose (EC) and polyethylene glycol (PEG) small-molecule solutions in the core sheath to simultaneously load ibuprofen (IBU) through an improved coaxial ESP. The prepared engineered spindles-on-a-string (SOS) nanofiber hybrid possesses typical controlled drug release properties. When the SOS structural mixture contacts water, the hydrophilic small molecule PEG rapidly dissolves to release IBU, while the hydrophobic EC drug on the other side is continuously released in a slow-release form (Figure 6A) [113]. This precise controlled release means providing the patient with the desired drug environment in the first instance, followed by a prolonged slow release to provide effective blood levels [114,115,116,117].

Facing the complex living system of organisms, hydrophobic drugs confront barriers in the delivery pathway. The use of NPs as carriers can improve the solubility of hydrophobic drugs. Simultaneously, the Enhanced Permeability and Retention (EPR) effect of NPs can stabilize tumor aggregation [115]. These benefits steadily improve the use and efficacy of hydrophobic drugs [117]. Recently, Xu et al. prepared CUR/CUR@MSNs-NFs by ESP by loading curcumin (CUR) in SiO_2_NPs. The scaffolds exhibited excellent biphasic release, which released 54% in the first 3 days and all in 35 days [118]. In addition, by modifying the SiO_2_NPs, the amine-functionalized MSNs are hydrophilic and positively charged, resulting in a more prominent anti-cancer effect [119]. Li and co-workers used amphiphilic lignin nanoparticles (LNPs) loaded with the anticancer drug paclitaxel (PLNPs) encapsulated in PVA/PVP nanofibers, which exhibited a 59% initial abrupt release compared to the rapid abrupt release of PLNPs (Figure 6B) [115]. In the same manner, adriamycin DOX was inserted into layered nanohydroxyapatite LHAp (DOX@LHAp) to blend it with PLGA. For obtaining nanofibers, DOX was loaded with 2D layered material. The release showed significant attenuation at the initial stage of release and subsequent strong prolonged release. The in vitro release showed excellent controlled release ability (Figure 6C) [116]. However, the relevant tests are still in the laboratory stage, and the in vivo situation is more complex because there are many influencing factors. Therefore, the in vivo drug release situation needs further research.

#### 3.1.2. Dual-Drug Biphasic Approach

Inside the face of this more complicated therapeutic environment, the role of a single drug is considerably limited. To treat this shortcoming, the dual-drug biphasic approach is adopted. The ES process is used to load various drugs onto different materials. In a dual-drug biphasic release, core-shell and Janus composite fibers are frequently employed [120]. The existence of beads on a string in the ESP is frequently unloved, with most people believing that smooth fibers and uniform particles are the desired result. Nevertheless, according to the Janus beaded fibers prepared by Li and co-workers through a homemade eccentric spinning head with a hydrophilic polymer PVP containing the MB model drug and a hydrophobic polymer EC containing ketoprofen (KET) on other side, the beads on a string have more advantages in the drug release phase compared to the prepared Janus nanofibers (Figure 7). The drug is evenly enclosed in the fibers throughout the stretching process, forming a good compartment. Because of the low viscosity of the solution during the stretching phase, the beaded structure generates an unstable jet during electrospinning. This phenomenon causes polymer aggregation to generate discrete bumps. The hydrophilic side of Janus beads releases MB quickly, whereas the hydrophobic side takes longer to release ketoprofen (IPU). This facilitates a dual-drug controlled release [121]. Then, encapsulating a nanoparticle-loaded drug in a nanofiber loaded with another drug may yield pleasantly exciting results in a controlled drug release through the difference of their properties [116]. Gupta et al. formed core-shell fibers by loading kaempferol in biodegradable albumin nanoparticles that were monolithically encapsulated in PCL loaded with dexamethasone [122]. After a weak burst release at 24 h, a sustained slow release of 15 days is possible, allowing the prolonged simultaneous action of both drugs to synergistically promote osteogenesis. Furthermore, single encapsulation of kaempferol-loaded albumin nanoparticles in PCL nanofibers with in vitro dissolution of the drug revealed a decrease in drug release rate from both composite fibers, but this is a gratifying result, where the presence of nanoparticles reduces the continuity of the fibers, resulting in a decrease in the drug release rate.

#### 3.1.3. Smart Response Drug Delivery

Characteristics, such as site-specific rather than systemic action, controlled drug release and intelligent responsive release, need to be satisfied for effective drug delivery systems [123,124,125]. The advantages of NNHs are evident in blending functionalized nanoparticles within electrospun fibers, which can facilitate the combination of material benefits to achieve the desired properties [126]. Liu et al. developed a fiber surface growth particle morphology fiber through secondary growth of an MOF material by adjusting different concentrations of ligands (2-MIM) and the reaction time. The number of fiber surface particles (ZIF-8) grown produced differences in drug release analysis. When the concentration of ligands (2-MIM) is increased, the more ZIF-8 particles grow on the fiber, leading to a decrease in the amount of drugs released. The presence of nanoparticles significantly reduced the drug release capacity. After 72 h of dissolution experiments, 89% of APS was released at pH = 5.5, and only 40% of APS was released at pH = 7.2. The relative extremely slow rate obviously depends on the pH value (Figure 8B) [127]. Croitor et al. prepared NNHs by a single-step blending process of 10% PLA with different masses of GO stirred well and by ESP [120]. The surface of the fiber is flat and smooth, and the diameter of fiber decreases significantly after loading GO nanoparticles. GO has superior electrochemical properties and can respond rapidly under external electric field stimulation. Through an in vitro drug dissolution test, the drug release capacity at 10HZ PLA/GO/Q is about 8000 times more than that without external stimulation (Figure 8C). In the treatment process, the ability to achieve an intelligent response to the characteristics of drug release through the modulation of electrical signals alone is undoubtedly a groundbreaking innovation. However, in the drug delivery process, it is necessary to provide the appropriate release rate. Too fast a release of drugs can easily reach the peak, biological activity can become to high or a certain degree of toxicity can even be produced. How to precisely regulate the ability to release drugs through changes in electrical signals is still a challenge [128].

For the same purpose, Banerjee et al. generated a PCL composite fiber containing superparamagnetic iron oxide nanoparticles (SPIONs). The NPs can carry rhodamine B model drugs. Under external stimulation, the NPs generate magnetically. Furthermore, the NPs create thermal energy as a result of Néel–Brownian relaxation [129]. The morphology of the composite under human-acceptable laser irradiation showed a molten state compared to pure PCL electrospun fibers with no significant change, which was attributed to the second effect of SPIONs (Figure 8A). In vitro assays showed that the release of rhodamine B from RF-EMF exposure showed a linear increase at a certain time, reaching 40% release after the fifth activations. Controllability is perfectly illustrated. Nevertheless, one should be aware that the magnetocaloric effect of SPIONs is complementary to thermosensitive polymers, but the generated high heat may affect the performance of the polymers as well as the biological activity of the heat-insensitive drugs [130,131].

### 3.2. Antibacterial

Recently, with the abuse of antibiotics, different types of drug-resistant bacteria emerged [39,132,133,134], and the combination of different types of materials based on NNHs can also generate excellent antibacterial ability against drug-resistant bacteria (Table 2). The preparation of NNHs by loading the bactericidal nanoparticles in the precursor solution begins with the mixed uniform solution, which is prepared into fibers by ESP. Mainstream bactericidal nanoparticles mainly include metal (AuNPs [135,136], AgNPs [137,138]) and metal oxides (ZnO [139], CuO [140] and TiO_2_ [141]), etc. The accumulation of AgNPs in mitochondria leads to mitochondrial dysfunction. Additionally, AgNPs disrupt the DNA structure, resulting in non-replication and effectively suppressing bacterial multiplication. These two fundamental factors lead AgNPs to possess potent bactericidal potential [142]. Li et al. manufactured PCL nanofibers loaded with AgNPs and cisplatin (DDP), which can be used to prevent airway inflammation and resist granulation tissue proliferation. In comparison with the control group, the PCL-DDP-AgNPs scaffold showed no adhesion to rabbit peritracheal tissues, as shown in Figure 9D, while the coated plate had a superior antibacterial effect [137]. Yang and co-workers prepared Janus nanofibers PVP-CIP/EC-AgNPs by loading AgNPs on the EC side through bide-by-side electrospinning (Figure 9A). The Janus fibers from the disc-diffusion experiment showed a larger circle of inhibition against *S. aureus* and *E. coli* than fibers loaded with a single antimicrobial agent, which may be due to the simultaneous action of AgNPs and CIP, steadily increasing the antimicrobial capacity [132]. AgNPs are easily aggregated in polymer fibers, which affects the mechanical properties of the fibers and significantly compromises the antibacterial effect. Bakhsheshi-Rad et al. doped GO/AgNPs into PLLA fibers. They deposited composite fibers onto a bio-implant Mg alloy. SEM images of composite fibers co-cultured with bacteria revealed considerable bacterial cell membrane disruption (Figure 9B). When compared to PLLA nanofibers, GO/AgNPs/PLLA efficiently inhibits bacterial proliferation [133]. This strategy provides new ideas for surface modification of metallic biomaterials, but high concentrations of AgNPs possess a degree of toxicity to normal cells. There is no presence of Ag elements in the organism. For example, if they are mostly deposited inside the liver, they will cause irreversible damage to the organism. Therefore, how to effectively and precisely control the amount of AgNPs released deserves researchers’ deep thinking [143]. There is a similar situation for AuNPs. Ibrahim et al. employed carboxymethyl chitosan (CMCS) as a green reducing agent for AuNPs. AuNPs were encased in PVA nanofibers. This technique reduces the cytotoxicity of AuNPs generated by the chemical reductant method while efficiently preventing bacterial growth. It signifies that safety has been improved further (Figure 9C) [135].

### 3.3. Tissue Engineering

Simulating the structure and composition of the extracellular matrix (ECM) is a method to provide suitable conditions for cell adhesion, differentiation and proliferation. Researchers have found that electrospun fibers are identical in properties to natural tissues through the intersection of biological, medical and nano-engineering technologies [149,150] while offering high porosity. Electrospinning (ES) is considered one of the most eligible technologies for tissue engineering, providing scaffolds for tissues to mimic ECM composition and deliver Biofactors, thus promoting the growth of new tissues [151].

We will now discuss the functionalization of electrospun fibers by co-blending nanoparticles in the spinning system. Lu and colleagues used SiNPs deposited on the surface of PLLA electrospun fibers, which effectively enhanced the mechanical properties and hydrophilic ability of the composite membrane, leading to better biocompatibility and promoting cell adhesion and proliferation [152]. Orthopedic clinical challenges are surgical risks of bone loss aptamers and pathogenic infections. Placing autografts or an Allograft is the best clinical option to combat bone loss but with immune rejection problems and infection risks [153]. Following the development of NNHs, electrospun fiber tissue engineering scaffolds can be fabricated from nanofibers and several nanoparticle hybrids, which mainly include: inorganic nanoparticles (nanohydroxyapatite (nHA) [154], calcium phosphate (CaPs) [155], Molybdenum Disulfid (MoS_2_) [156], Cerium dioxide (CeO_2_) [63] and Magnesium oxide (MgO) [157]) and metal nanoparticles (gold nanoparticles (GNPs) [158]), etc.

Nanohydroxyapatite (nHA) enhances the osseointegration and osteoconductive properties of hard tissues by embedding itself in the collagen matrix of hard tissues. Song and co-workers spun NELL-1 functionalized chitosan nanoparticles (NNPs) and nHA in PCL fibers (Figure 10A). By incorporating the NPs into PCL electrospun fibers, the authors clearly increased the fiber diameter and demonstrated that nHA and NNPs enhanced cell adhesion sites. nHA presence compared to unmodified nanofibers MC3T3-E1 cell proliferation and differentiation were enhanced [159]. Liu and colleagues prepared hydrogel fibrous scaffolds containing gelatin-methacryloyl (GelMA) and calcium phosphate nano-particles (CaPs). After 14 days of incubation in SBF, significant mineralized nodules developed on the surface of CaPs@GelMA-F. Faster Ca^2+^ deposition in the early stage can significantly boost the development of calcified nodules later on (Figure 10B). Moreover, bone defects are accompanied by rupture of blood vessels. CaPs@GelMA-F cells were co-cultured with HUVECs cells, resulting in pictures of dense vascular network topology. CaPs@GelMA-F is also very effective in promoting angiogenesis [160].

Because of its unique physicochemical properties, where S is an important component of many amino acids in living organisms, MoS_2_ is also an essential trace element in the human body [141]. Researchers found that doping MoS_2_, a two-dimensional material, with nanofibers can effectively promote cell proliferation and maintain cell viability of BMSCs, and at the same time, the higher the MoS_2_ content, the stronger the osteogenic ability [161,162]. Furthermore, Ma et al. found that the NIR photothermal properties possessed by MoS_2_ can respond quickly, reaching the optimal temperature required for osteogenesis (40.5 ± 0.5 °C) in 30 s with 808 nm NIR irradiation, as shown in Figure 10C. The BV/TV capability under NIR irradiation reached 41.41 ± 0.52% [156].

In addition to the various applications in osteogenic tissues, NNHs also have a positive role for other tissue engineering formation. Neural tissue engineering (NTE) is an emerging field. Material preparation requires excellent biocompatibility, certain mechanical properties and good permeability to oxygen and nutrients, etc. [163]. Chen and co-workers demonstrated the combination of melatonin MLT and Fe_3_O_4_-MNP with PCL to prepare an outer-middle-internal triple-layer structural scaffold of PCL, Fe_3_O_4_-MNPs/PC and MLT/PCL, with MLT effectively reducing oxidative loss. The results revealed that the scaffold had a medullary axon diameter (3.30 μm) similar to the autograft group, which significantly promoted neuronal axon growth (Figure 11D) [164]. Myocardial infarction is one of the more common diseases in humans. Myocardial cells are gradually replaced by scar tissue after necrosis, leading to heart failure and arrhythmias caused by changes in electrophysiological properties, so it is most important for cardiac tissue engineering to have excellent elasticity, high electrical conductivity and promote the growth of myocardial cells [165]. Zhao et al. employed the ES process to create carbon nanotube/silk protein (CNT/silk) scaffolds. The majority of carbon nanotubes are contained within the fibers and dispersed along the fiber development direction. Compared to monospun silk protein, the elongation at the break of the composite fiber is up to 200% or more, and the tensile strength reaches 5.0 Mpa (Figure 11C). The CNT/silk stent’s excellent electrical conductivity implies that it is efficient in avoiding arrhythmia (Figure 11B) [166].

The human immune system’s first line of defense is the skin. Skin injury is common in everyday life. Wound dressings to meet the demands of skin tissue engineering are scarce. A great wound dressing must not only support ECM regeneration but also protect the skin from exogenous microorganisms [168]. NNHs fulfill the need for skin tissue engineering owing to their versatility. Janus nanofibers were loaded with AgNPS and RCSPs to impart antibacterial ability and bioactivity to electrospun fibers. In the present work, interestingly, Janus nanofibers can be obtained based on the uniaxial electrostatic spinning process through the presence of phase separation of PCL and PVP in a solution. AgNPS in the free state is very susceptible to phagocytosis by normal cells due to its fine particles and exists a certain degree of cytotoxicity, and RCSPs-Ag nanofibers in this study had only 78.86% viability on NIH 3T3 cells [169]. The team designed a sandwich wound dressing with AgNPs loaded with an intermediate layer of nanofibers to both impede the invasion of exogenous microorganisms and avoid cytotoxicity (Figure 11A) [167].

## 4. The Present Challenges of NNHs in Medical Applications

Nanomedicine technology is constantly evolving. The use of functionalized nanofibers and nanoparticle hybrids (NNHs) in medicine has shown encouraging effects. However, there are a number of challenges that must be surmounted: (1) Most nanoparticles and electrospun fibers necessitate the use of organic solvents in their fabrication. Organic solvent residues make the composites less biocompatible. The green chemistry of preparation should be considered. (2) The large-scale preparation of NNHs remains a challenge in the face of industrialization needs. Multi-needle collaborative electrospinning has issues with jet interactions, needle clogging and cleaning difficulties. Needle-free electrospinning method initially allows the production of large quantities of nanofibers, but they are exceedingly inhomogeneously dispersed. (3) Nanoparticles are sporadically arranged in nanofibers, and NNHs have weak mechanical properties. This makes them more fragile as tissue scaffolds for tissue engineering applications. It is vital to investigate the “bridge” that improves the interaction of NPs with the substrate. (4) Precisely controlling nanoparticle alignment in fibers is a huge difficulty. This has the potential to have a significant impact on the electron transport efficiency of NNHs in electrically sensitive drug release. This has a detrimental impact on the biosensor as well. (5) The studies on NNHs in drug release were carried out in a more ideal environment. When confronted with the organism’s complicated physiological environment, more research is needed to determine how to precisely control the drug’s initial burst release and duration of activity.

## 5. Conclusions and Outlook

In summary, combining current advanced technologies, nanofibers and nanoparticle hybrids (NNHs) have a wide range of applications in the medical field, especially in drug release, antimicrobial and tissue engineering. The electrospinning process (ESP) developed NNHs provides a convenient way to load nanoparticles (NPs) in nanofibers (NFs) in a single-step process; however, both dispersion of NPs in a polymer solution and interfacial interactions directly affect the structure and properties of NNHs, which can be sprayed directly on fibers by physical techniques (magnetron sputtering, plasma, electrospray, etc.) or chemical synthesis based on nanoparticle methods (in situ synthesis, hydrothermal-assisted, calcination, etc.) to grow NPs inside NFs.

Combining novel NPs has been attempted. Hypercrosslinked polymers with NFs are still in a preliminary stage, which is based on Friedel–Crafts reaction [170,171]. The one-step synthesis of NPs possesses controlled morphology [172], a specific surface area as high as 4000 m^2^/g, excellent biocompatibility and high porosity often used for drug delivery [173]. Combining both in the medical field is beneficial for the development of new materials.

Clinical applications frequently demand scaffolds to be multifunctional, not only to enhance cell reproduction but also to suit specific needs [174]. Scaffolds for tissue engineering must also have regenerative properties, such as the capacity to promote blood vessel formation or heal torn tissues [175]. More functional NPs will be placed into NFs in the future to create novel biomaterials based on newer ES technology iterations that combine more functional particles with multi-fluid technology (three-fluid or even four-fluid ESP). The materials’ synergy may result in a new form of superfunctional composite. In the future, humans will have to deal with complicated illness challenges. Personalized custom brackets must be carefully considered. The mechanical characteristics of NNHs have remained a source of contention. Three-dimensional printing technology is being used with ESP to create an artificial intervertebral disc scaffold [176]. The mechanical properties of the stent were all above the normal human threshold. Any use of NNHs in conjunction with 3D printing technology is a potential trend. Materials are highly demanding for use in living organisms. The high compression modulus of the human bionic scaffold can readily injure neighboring tissues. While the modulus is too low, it is impossible to play the bracket role [52]. Computational simulations are used to determine the best solution, and as a result, a new sector of material design for NNHs will emerge [177]. Needle-free ESP offers a novel approach to nanofiber preparation on a large scale [178,179]. The needle-free ESP used to prepare NNHs on a large scale is expected to be commercialized soon.

## Figures and Tables

**Figure 1 polymers-14-00351-f001:**
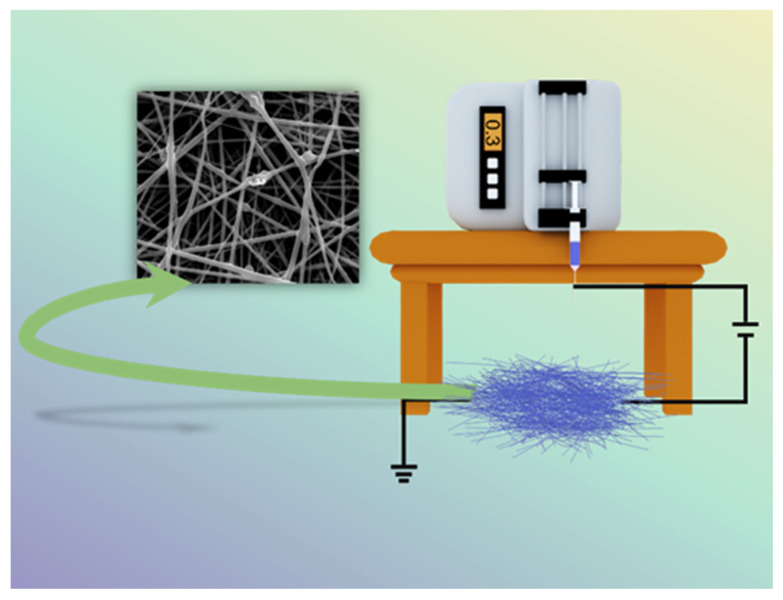
Schematic diagram of electrospinning setup.

**Figure 2 polymers-14-00351-f002:**
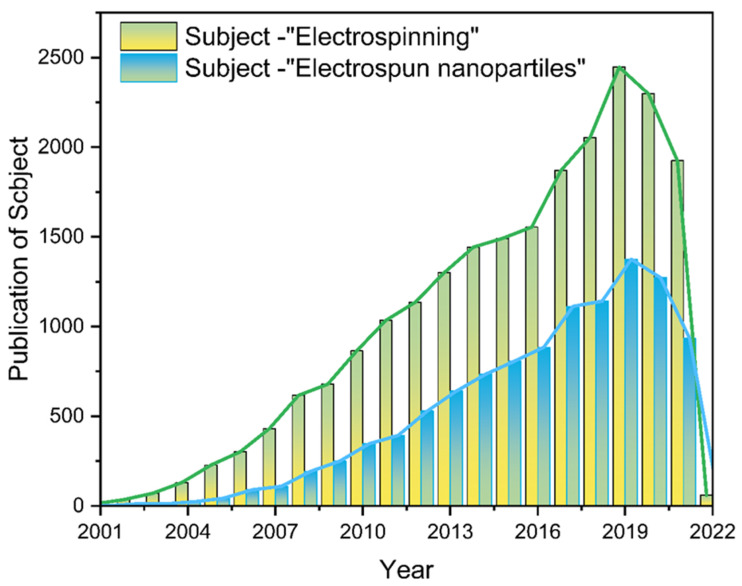
The literature search statistics of “electrostatic spinning” and “electrospun nanoparticles” on the “Web of Science” platform, respectively.

**Figure 3 polymers-14-00351-f003:**
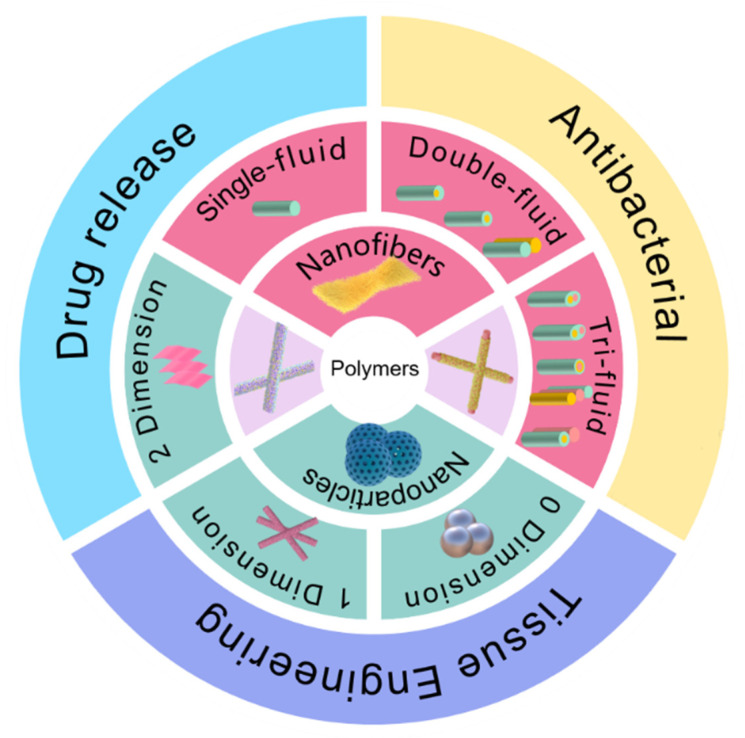
Electrospun fiber and NPs structures and their hybrids in medical direction.

**Figure 4 polymers-14-00351-f004:**
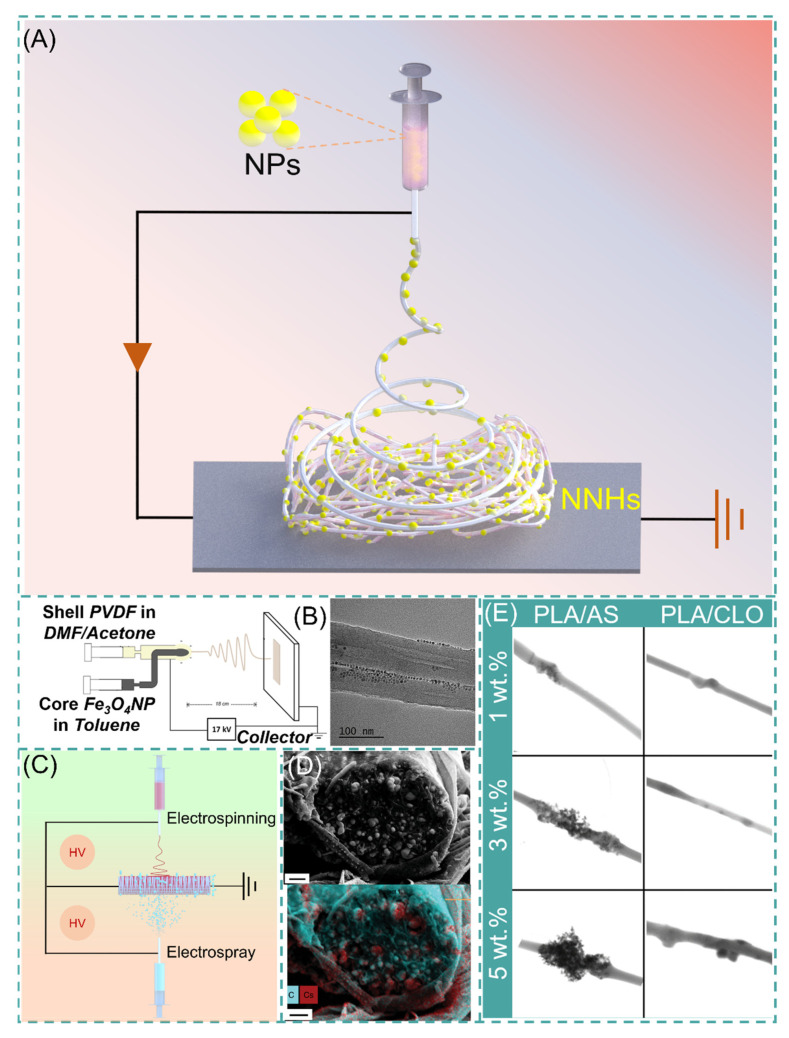
(**A**) Simple preparation of NNHs based on the single-step process. (**B**) Schematic diagram of coaxial preparation of NNHs and Transmission Electron Microscopy (TEM), reprinted with permission from Ref. [84]. Copyright 2021 John Wiley and Sons. (**C**) Schematic diagram of simultaneous electrospinning and electrospraying. (**D**) Scanning Electron Microscopy (SEM) images of CDP-PVP-PANI fiber cross-sections and Energy dispersive X-Ray spectroscopy map. Reprinted from Ref. [85]. (**E**) Scanning Transmission Electron Microscopy (STEM) images of PVA/AS or PVA/CLO fibers with different particle concentrations, Reprinted from Ref. [82].

**Figure 5 polymers-14-00351-f005:**
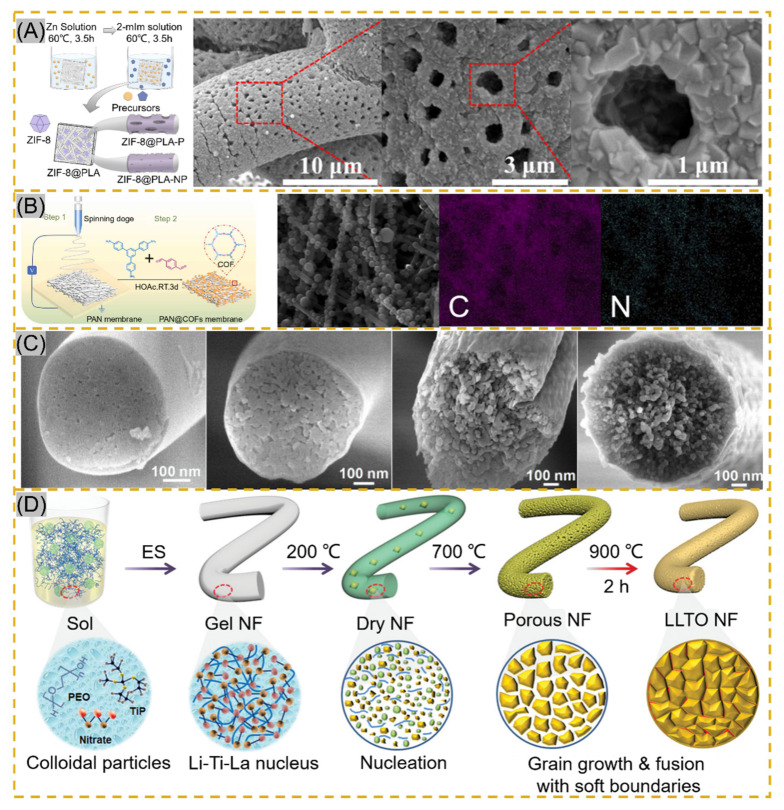
(**A**) Schematic of in situ ZIF-8 growth on PLA fibers and Scanning Electron Microscopy (STEM) images of ZIF@PLA-P, reprinted with permission from Ref. [95]. Copyright 2021 ACS Publications. (**B**) Schematic of PAN@COF synthesis and EDS mapping images reprinted with permission from Ref. [91]. Copyright 2022 Elsevier. (**C**) Transmission Electron Microscopy images of MBNM films prepared based on different contents of PAA, reprinted with permission from Ref. [96]. Copyright 2021 Elsevier. (**D**) Schematic of LLTO NF process prepared by electrospinning followed by calcination, reprinted with permission from Ref. [97]. Copyright 2021 John Wiley and Sons.

**Figure 6 polymers-14-00351-f006:**
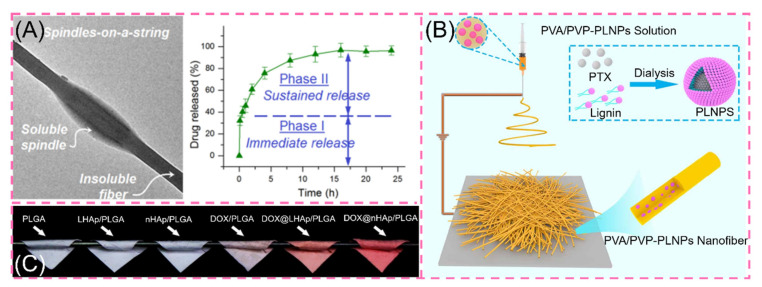
(**A**) Transmission electron microscopy images of engineered spindles-on-a-string (SOS) and in vitro drug dissolution curves reprinted with permission from Ref. [113]. Copyright 2021 Springer Nature. (**B**) Preparation process of PVA/PVP-PLNPs nanofiber membrane reprinted from Ref. [114]. (**C**) Digital photographs of various fiber scaffolds reprinted with permission from Ref. [116]. Copyright 2019 Elsevier.

**Figure 7 polymers-14-00351-f007:**
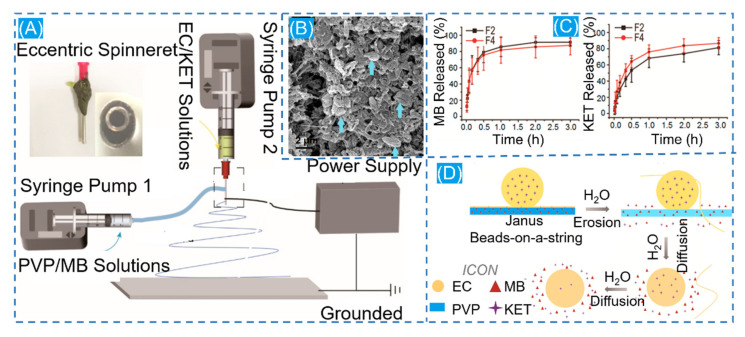
(**A**) Schematic diagram of Janus nanofiber preparation. (**B**) Schematic diagram of scanning electron microscopy of residual ECNPs after solubilization. (**C**) In vitro drug dissolution profiles for MB and KET. (**D**) Diagram of the drug release mechanism. Reprinted from Ref. [121].

**Figure 8 polymers-14-00351-f008:**
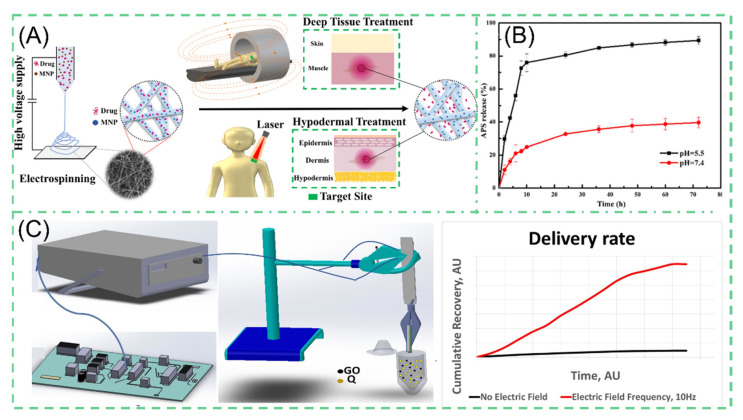
(**A**) Schematic diagram of the preparation process of magneto-thermal responsive nanofiber MSP reprinted from Ref. [129]. (**B**) Drug release profile of PH-responsive NNHs reprinted with permission from Ref. [127]. Copyright 2020 Springer Nature. (**C**) Drug release from electrically stimulated PCL/GO/Q composite nanofiber under 10 HZ electrical stimulation reprinted from Ref. [120].

**Figure 9 polymers-14-00351-f009:**
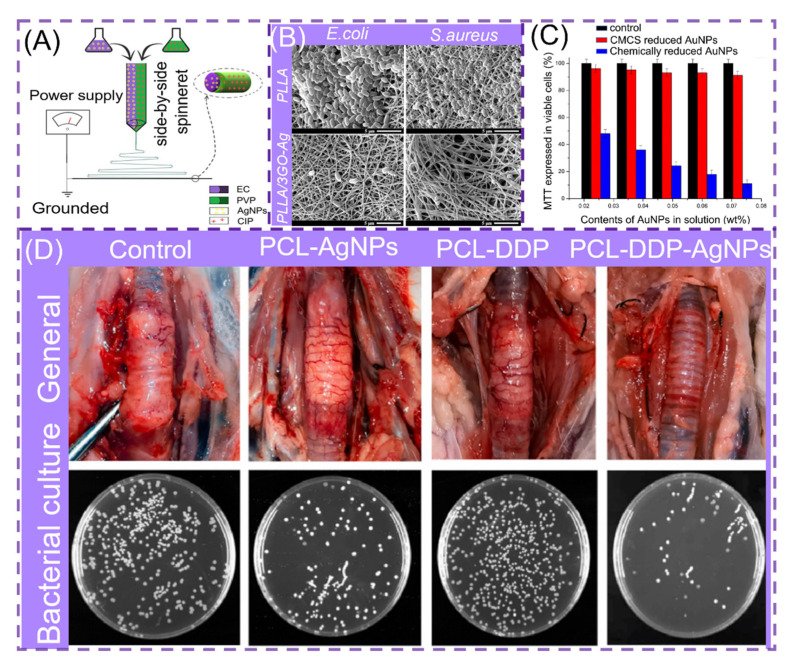
(**A**) Process diagram for the preparation of parallel structured NNHs, reprinted with permission from Ref. [132]. Copyright 2020 Elsevier. (**B**) Scanning electron microscopy (SEM) images of bacteria cultured on PLLA and PLLA/3GO-Ag, reprinted with permission from Ref. [133]. Copyright 2020 Elsevier. (**C**) MTT method to measure the cellular activity of various materials on epidermal cells, reprinted with permission from Ref. [135]. Copyright 2020 Elsevier. (**D**) Optical images of bronchial stents on rabbit trachea and assessment of bacterial inhibition by plate count method, reprinted from Ref. [137].

**Figure 10 polymers-14-00351-f010:**
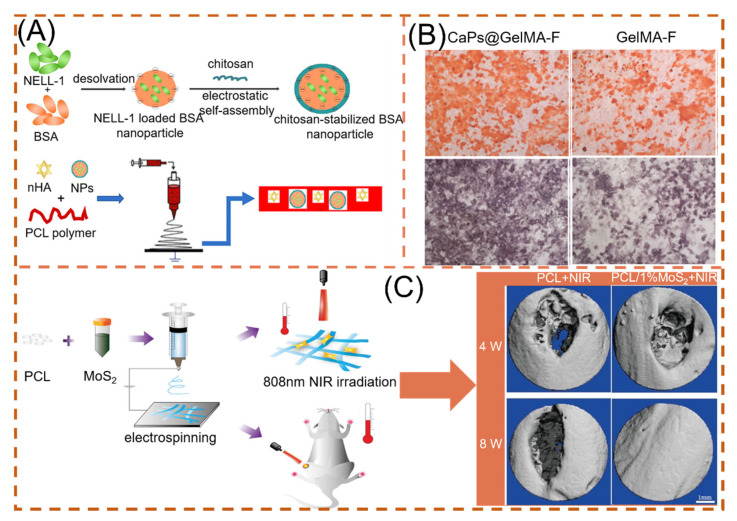
NNHs for promoting bone tissue growth, (**A**) Schematic preparation of loaded dual nanoparticle electrospun scaffolds, reprinted from Ref. [159]. (**B**) Osteogenic ability of loaded CaPs with unloaded electrospun fibers (after alizarin red staining and), reprinted with permission from Ref. [160]. Copyright 2020 Elsevier. (**C**) PCL and MoS_2_ co-blended preparation of NNHs. Schematic diagram and micro-CT imaging with and without MoS_2_ under photothermal conditions, Reprinted with permission from Ref. [156]. Copyright 2021 John Wiley and Sons.

**Figure 11 polymers-14-00351-f011:**
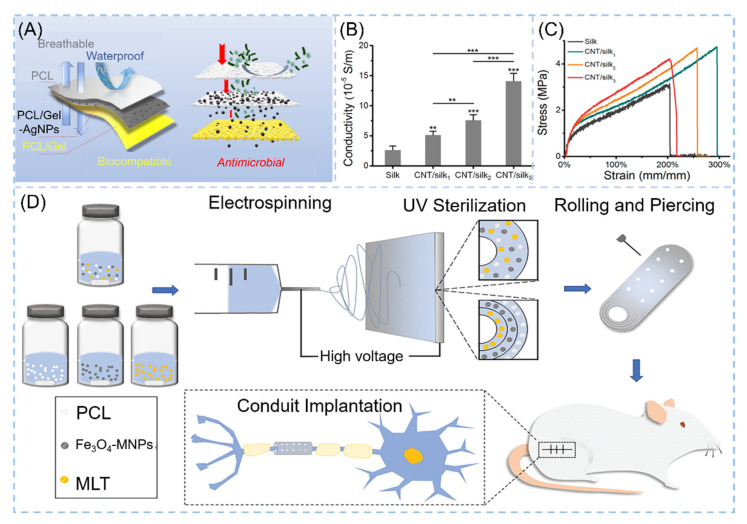
NNHs for other tissue engineering applications. (**A**) Schematic diagram for skin tissue engineering, Reprinted with permission from Ref. [167]. Copyright 2021 Elsevier. Tensile curves of CNT/silk scaffolds (**B**) for cardiac tissue engineering and (**C**) electrical conductivity, Reprinted with permission from Ref. [164]. Copyright 2020 John Wiley and Sons. (**D**) Process flow diagram for neural tissue engineering preparation, Reprinted with permission from ref. [164]. Copyright 2020 John Wiley and Sons.

**Table 1 polymers-14-00351-t001:** Effect of control parameters on the morphology of electrospun fibers.

	Influence Factors	Influence Results	Reason for Influence	Ref.
System parameters	Polymer concentration	The higher the concentration, the coarser the fiber	As polymer concentration or molecular weight increases, so does solution viscosity. Greater entanglement between molecule chains and increased intermolecular Coulomb forces result from this condition. As a result, the fiber diameter expands.	[35]
	The molecular weight of polymers	The higher the molecular weight, the thicker the fiber		[36,37]
	Surface tension	The higher the surface tension, the finer the fiber	The droplet’s surface tension rises, and the jet must expend more energy to offset this negative effect. The speed of the jet slows down, requiring more time to stretch the fibers. As a result, the fiber diameter decreases.	[38]
	Conductivity	Conductivity increases within a reasonable range; fiber diameter decreases and increases again; fiber diameter is not controllable	The charge accumulates on the surface of the jet when the conductivity is increased. The fibers stretch more quickly in this state. As a result, the diameter of the fiber is lowered. The coulombic repulsion at the jet interface is intensified when the solution conductivity is raised further. Uncontrollable fiber diameter distribution results from the unstable bending whip effect.	[39,40,41]
Process parameters	Voltage	The fiber diameter decreases with higher voltage and increases with higher voltage	As the voltage rises, the charge density on the jet’s surface rises in accumulation. The circumstance may lead to a significant effect of jet stretching. Therefore, the fiber diameter decreases. The flow rate at the spinneret, on the other hand, increases as the voltage is raised more. Instead, the diameter of the fiber rises.	[42]
	Flow rate	The flow rate increases; the fiber diameter increases; and further increases may result in droplets	The solution at the spinneret rises as the flow rate increases. This condition causes the fiber’s diameter to thicken. When the flow rate is too fast, the solution’s gravity causes it to trickle straight down.	[43,44]
	Receiving distance	Acceptance distance enlarges and fiber diameter reduces	The additional receiving distance gives the jet more time to extend. The fiber’s diameter shrinks in this circumstance.	[45]
Environmental factors	Temperature	Within a reasonable range, the fiber diameter decreases as the temperature increases	The temperature has the greatest influence on the viscosity of the solution. The viscosity of the solution reduces as the temperature rises. The intermolecular Coulomb force is lessened in this scenario.	[46]
	Humidity	Humidity increases and grooves appear on the fiber surface	When humidity is too high, fiber production is accelerated. Water droplet condensation on the fiber surface. Wrinkles occur on the surface of the fibers as a result of this process.	[47]

**Table 2 polymers-14-00351-t002:** Examples of NNHs used as antimicrobial agents.

Polymers	NPs	NNHs	Preparation Methods	Bacterial Strains	Evaluation Methodology	Antibacterial Ability	Ref.
PVA/CS	CuNPs	PVA/CS/Cu	Co-blending	*S. Aureus* (ATCC 25923); *B. cereus* (ATC 11788); *E. coli* (ATCC 35218); *P. aeruginosa* (ATCC 49189)	Antibacterial Circle	The size of the inhibition circle is: *S. Aureus* (15.6 ± 1.1 mm); *B. cereus* (29.6 ± 0.42 mm); *E. coli* (13.3 ± 0.8 mm); *P. aeruginosa* (10 ± 1 mm)	[144]
GEL/PCL/P(DMC-AMA)	nHAP	JGM	Co-blending	*E. Coli* and *S. Aureus*	CFU Counting	The bacterial viability of *S. aureus* after 6 h was 0.1%.	[145]
Starch	AgNPs	starch/AgNPs	In-situ synthesis	*E. Coli* (ATCC 35218); *S. Aureus* (ATCC 29213)	Disc Diffusion-8mm	*E. coli* (9.7 mm); *S. Aureus* (10.2 mm)	[96]
PMMA	ZnO nanorods/AgNPs	PMMA/ZnO-Ag NF	Co-blending, in situ synthesis	*E. coli* (ATCC 25922); *S. aureus* (ATCC 25923)	Disc Diffusion-6mm	*E. coli* (7–17 mm); *S. Aureus* (8.5–18.5 mm)	[81]
CH/PEO	8Ce-BG	CH-PEO-(8Ce-BG)	Co-blending	*E. Coli* and *S. Aureus*	Flat Counting Method	*E. coli* activity was only 55.3%	[146]
PLLA	GO-Ag	PLLA-GO-AgNPs	Co-blending	*S. aureus* (ATCC 12600); *E. coli* (ATCC 9637)	Antibacterial Circle	3.01 mm–4.62 mm	[133]
PVP K90/EC	CIP/AgNPs	PVP-CIP//EC-AgNPs	Co-blending	*S. aureus* (ATCC 27853); *E. coli* (ATCC 25922)	Antibacterial Circle	24 h, *E. coli* (17.8 ± 0.6mm mm); *S. Aureus* (21.9 ± 0.6 mm)	[132]
PVA	ZnO	PVA/ZnO	Self-assembly	*S. Aureus* (ATCC25923); *E. coli* (ATCC25922)	MIC method	*E. coli* (62.5 μg/mL); *S. Aureus* (250 μg/mL)	[147]
PLGA/SF	ZnO	PSZ	Co-blending	*E. Coli* and *S. Aureus*	turbidity measurement method	PSZ antibacterial activity against *S. Aureus*: 45.1–100%	[148]

## Data Availability

Not applicable.

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
