# Peer review of "Polymer-Based Nanofiber–Nanoparticle Hybrids and Their Medical Applications"

_polymers, 2022, doi:10.3390/polym14020351_

Round 1
Reviewer 1 Report
This is an interesting article. Following very minor corrections are needed:
-Authors should make clear how the work is different from others
-Authors should try to draw their own figures
-Future perspectives should be elaborated
-Some of the relevant references may be cited such as:
Materials Today Chemistry 17, 100301(2020); Polymers 2022, 14(1), 89; Carbohydrate Polymers 272, 118491(2021)
Reviewer 2 Report
Greetings, Editor thank you for providing me with the opportunity to review the article. I reviewed the article with title ``Polymer-based Nanofiber-Nanoparticle Hybrids and Their Medical Applications ``. The theme of the article is very interesting and promising in the field. Overall, the structure and content of article is acceptable for the polymer. I am pleased to send you major level comments because there are some serious flaws which need to be correct before publication. The manuscript can be accepted for publication after modification. Please consider these suggestions and comments as listed below.
- The first sentence of abstract is wired. The sentence is too long. There are few grammatical mistakes please revised the abstract very carefully. Introduce one line in the beginning about study introduction.
- Research gap should be delivered on more clear way with directed necessity for the conducted research work.
- The novelty of the work must be clearly addressed and discussed.
- Each reference needs to be properly addressed. Please revise your paper accordingly since same issue occurs on several spots in the paper.
- Introduction section must be written on more quality way. Please update the reference from 2015-2021 only. In introduction Line 32 need this reference to cite- Safian, M.T.; Umar, K.; Parveen, T.; Yaqoob, A.A.; Ibrahim, M.N.M. Chapter Eight-Biomedical applications of smart polymer composites. In Smart Polymer Nanocomposites: Biomedical and Environmental Applications; Woodhead Publishing Series in Composites Science and Engineering; Elsevier Inc.: Cambridge, MA, USA, 2021; pp. 183–204.
- Page 1 Line 33 in introduction please cite this reference to support your statement.- Yaqoob AA, Safian MT, Rashid M, Parveen T, Umar K, Ibrahim MN. Introduction of smart polymer nanocomposites. In Smart Polymer Nanocomposites 2021 Jan 1 (pp. 1-25). Woodhead Publishing.
- The main objective of the work must be written on the more clear and more concise way at the end of introduction section.
- Line 432 need another closely relevant reference. Please cite this Recent advances in metal decorated nanomaterials and their various biological applications: a review, with your reference [39]. Please do not remove your reference [39].
- Please remove the word ``WE`` from the manuscript.
- Please follow the journal guidelines to prepare the article. Please check reference carefully.
- Please add one section about the Present Challenges of the field.
- Conclusion section is missing some perspective related to the future research work, quantify main research findings.
- Major level English language should be carefully checked and carefully check for language typos. The sentence must be concise. It should not too long. Please revise your paper accordingly since same issue occurs on several spots in the paper.
As already mentioned, these are comments to improve the manuscript and not necessarily to down the quality of work, which is very good.
Reviewer 3 Report
Mingxin Zhang et al have reviewed Polymer-based Nanofiber-Nanoparticle Hybrids and Their Medical Applications. The draft needs extensive revision before it can be accepted for publication. Few suggestions are given below
1: The authors have used very long sentences, which has resulted in losing the meanings of the sentence. Instead they should divide long sentences into several short sentences that will make the draft more meaningful.
See
Similar to the development of pioneering research, the elec-trospinning process has stumbled through time. In the 16th century, William Gibert discovered in life that charged amber near water droplets form conical droplets out of which microdroplets are ejected........................................
The meaning of this sentences and the following sentence is not clear due to its length.
Similar long sentences are present at various places in the draft
Line 115-122 is a single sentence
Line 161-163
Line 202-206
Line 210-215
Line 279-282
2: Insert reference in Table 1 and rewrite the text in comprehensive way in column 4 with the heading 'reason for influence'
3: Heading of section 2.3 should be ' Formation of --------------' instead of forming
4: section 3.1.2 the heading should be Dual -Drug Biphasic approach
5: delete first 3 lines of conclusions, that contain the guide lines.
6: At various places in the draft the authors have cited the work of different researchers by mentioning the name of the team leader. For example ref. 106. Please note that a team leader is the corresponding author, not the first author. Therefore, use et. al or co-workers when referring the first author name
7: Figure 7 C is not clear
Round 2
Reviewer 2 Report
Dear Authors
I have reviewed again the manuscript and I think that it is ready for publication. Thank you for considering my suggestions.
Reviewer 3 Report
The authors have followed the suggestion for revision. The paper may now be accepted